# OpenReview forum: "Guiding LLM Decision-Making with Fairness Reward Models"
_NeurIPS.cc/2025/Conference — NeurIPS 2025 poster_

### Official Review · Reviewer_yvMq · 2025-06-24

**Clarity:** 3
**Significance:** 2
**Originality:** 2
**Rating:** 4
**Confidence:** 4

**Summary:**

This paper introduces a Fairness Reward Model for LLMs, focusing on improving the fairness of CoT in high-stakes decision-making tasks. The FRM is trained using weak supervision from LLM-annotated reasoning steps labeled as biased or unbiased. It is applied at inference time to reweight chain-of-thought (CoT) samples, thereby mitigating biases while preserving or even enhancing predictive accuracy. The effectiveness of the proposed method is demonstrated across three domains, demonstrating some improvements in fairness metrics such as equalized odds and equalized opportunity.

**Questions:**

Q1: How was the fairness prompting baseline constructed, and to what extent was it optimized for fairness?
The results suggest that the fairness prompting baseline underperforms significantly. It is unclear whether this reflects the inherent limitations of prompting or simply suboptimal prompt design.

Q2: Why were established fairness interventions (e.g., post-processing, adversarial debiasing, or fairness-aware finetuning) not included as baselines? This omission limits the strength of your empirical claims regarding the FRM’s effectiveness.

Q3: How do you justify using LLM-generated fairness labels, given their potential to encode the same biases they are meant to detect?
Although you report a 75% agreement rate with human judgments, the implications of this noise on model performance are not fully explored.

Q4: Why are only equalized odds and equalized opportunity considered, excluding individual fairness, counterfactual fairness, or worst group accuracy? The exclusive use of group fairness metrics may miss key dimensions of bias that are crucial in high-stakes domains.

**Ethical Concerns:**

["NO or VERY MINOR ethics concerns only"]

**Final Justification:**

Most of my concerns have been addressed, and I have updated my score accordingly.

**Limitations:**

yes

**Paper Formatting Concerns:**

No formatting issue

**Quality:**

3

**Strengths And Weaknesses:**

Strength:

(+): The proposal of a process-level fairness reward model that operates at the step-wise level of reasoning chains is novel and timely given the increasing deployment of LLMs in sensitive decision-making tasks.

(+): The model demonstrates notable generalization capabilities across tasks, domains, and even model families, which is a key challenge in fairness-oriented machine learning.

(+): The paper provides clear descriptions of model training, scoring, and decision aggregation mechanisms, including ablation studies that enhance reproducibility and transparency.


Weakness:

(-): The fairness annotations rely on GPT-4o-mini, whose own bias and opacity are acknowledged but insufficiently mitigated, raising concerns about the reliability of the supervision signal.

(-): The implementation and effectiveness of fairness prompting as a baseline is underexplained and appears suboptimal. As prompting is highly sensitive to design, stronger baselines (e.g., multi-shot or context-specific prompts) should be included.

(-): There is a lack of comparison with alternative fairness-aware techniques, including reweighting/post-processing methods or fairness-aware classifiers. The current evaluation primarily contrasts variants of their own architecture.

(-): Fairness metrics are presented as absolute values without normalized improvement percentages, making it difficult to assess relative gains especially across tasks with varied baseline disparities.

---

> ### Author Rebuttal · Authors · 2025-07-30
>
> We thank the reviewer for the time and effort taken to review our manuscript.  We are glad that the reviewer found our proposed algorithm to:
> - Be novel and timely
> - Demonstrate notable generalization capabilities, a key challenge in fairness-oriented machine learning
>
> Also, we appreciate that they found our presentation to be clear, transparent, and reproducible.
>
> Overall, we would like to emphasize that our proposed approach (and selection of baselines) were driven by several factors:
> 1. **Accuracy as first order concern**
> 2. **Generalization to new decisions tasks**
> 3. **Modularity**
>
> Our empirical results validate the potential for our approach to achieve both (1) and (2), whereas existing approaches primarily serve one goal or the other.  By unlocking this potential, we believe that our work makes a significant and original contribution to the field of LLM decision-making (as recognized by reviewers z2Cx and jRkp).
>
> Below we respond to particular concerns raised in the review in more detail.
>
> **[W2/Q1: Fairness prompting baselines]**
>
> The goal in designing our algorithm was to facilitate fair decisions while incurring a minimal (or no) accuracy tradeoff and without requiring any task-specific training data.  This reflects:
> 1. The practical need for accurate predictions
> 2. Legal realities around discrimination law in the US, where the existence of an equally accurate yet fairer model (i.e., a less discriminatory alternative) is a requirement for legal action against an organization accused of deploying discriminatory algorithms ([Black et al., 2024](https://arxiv.org/abs/2406.06817)).
> 3. The fact that many LLMs will be tasked with diverse one-off decisions for which no training data will be available (nor will training for each decision be practical).
>
> We contend (and find experimentally) that fairness prompting will fall short of delivering the required accuracy to be considered as a viable alternative to the most accurate available algorithm (noted in [Tamkin et al., 2023](https://arxiv.org/abs/2312.03689), and that (3) precludes the widespread use of fine-tuning base approaches. To avoid experimental bias in tuning each prompt, we used GPT-4o to produce both the base CoT prompt and the fairness prompt.  No prompts in our experiments received extensive tuning or were optimized for accuracy or fairness.  Additionally, we did not tune the fine-tuning hyperparameters for our FRM, instead adopting those of ([Snell et al., 2025](https://arxiv.org/abs/2408.03314); [Wang et al., 2024](https://arxiv.org/abs/2312.08935)).  These results support our claim that prompting is brittle and hurts accuracy. To further address these concerns, we have performed additional experiments on a set of stronger fairness prompting baselines.
>
> - **New experiments:** To further address these concerns, we have performed additional experiments on a set of stronger fairness prompting baselines. Using the COMPAS task, we systematically evaluated 10 original fairness prompts (written by GPT-4o) alongside the 7 prompts proposed by Tamkin et al. (2023) (https://arxiv.org/abs/2312.03689) to reduce discrimination in high-stakes language model applications. Prompts were tested with and without CoT and we additionally included a CoT@32 setting using the best-performing prompt from each group. To highlight the modularity of our approach, we also apply the FRM to combine the 32 CoTs produced using fairness prompts. We report equalized odds deviation (sum of absolute FPR and TPR differences) and equalized opportunity deviation (absolute TPR difference).
>
> - **New findings:** Consistent with our hypothesis and original findings, some prompts yielded reductions in equalized odds and equalized opportunity. However this came at the cost of significant reductions in accuracy. This tradeoff remained consistent across settings and accuracy was reduced further when we sampled 32 times with fairness prompts. In many cases, fairness was improved not through better reasoning but by predicting the same label for all inputs.  These results reinforce our core claim that reasoning-level supervision offers a more effective and robust fairness intervention than prompting when accuracy is a primary concern. Our findings echo concerns in prior work, including Tamkin et al. regarding output distortion from fairness prompting.
>
> - **Results**:
>
> | Setting                        | Avg. Accuracy | Min. Accuracy | Max. Accuracy | Avg. Eq. Odds Dev | Avg. Eq. Opp Dev |
> |-------------------------------|---------------|----------------|----------------|----------------|----------------|
> | 10 fairness prompts           | 46.6          | 27.8           | 56.6           | 16.0            | 6.7            |
> | 10 fairness prompts w/ CoT    | 56.7          | 53.6           | 60.2           | 16.2           | 4.0            |
> | 7 Tamkin prompts w/ CoT       | 58.1          | 54.6           | 59.4           | 17.5           | 3.8            |
>
> **Best Settings**:
>
> | Setting                                   | Accuracy | Eq. Odds Dev | Eq. Opp Dev |
> |------------------------------------------|----------|----------|----------|
> | Best fairness prompt w/ CoT (majority@32) | 57.2     | 19.7     | 4.7      |
> | Best fairness prompt w/ CoT (FRM@32)      | 63.6     | 16.6     | 4.1      |
> | Best Tamkin prompt w/ CoT (majority@32)   | 56.0     | 4.7      | 0.5      |
> | Best Tamkin prompt w/ CoT (FRM@32)        | 56.0     | 5.7      | 0.3      |
> | **FRM (Ours)**                            | **65.3** | **15.3**  | **7.6**  |
>
> **[W3/Q2: Other fairness-aware techniques]**
>
> We are not aware of any existing fairness method that is directly applicable to CoT reasoning with many samples, which is the focus of our work given our goal of achieving fairness under minimal disruptions to accuracy (i.e., creating a similarly accurate, yet more fair model).  While classic approaches like reweighting, adversarial debiasing and postprocessing adjustments are relevant in classification settings, they are largely orthogonal to our methods.
>
> Also, we emphasize that we offer a framework for improving fairness on decisions and tasks that were previously unseen during model training.  In many applications (including agentic ones), an LLM can be faced with many different and sometimes novel decisions, and it would not be practical or even feasible to fine-tune a different model for each decision.  Thus we perform our experiments in a setting where no training data is available, and benchmark against methods that do not require training data.
>
> Also, many such fairness techniques could be used in conjunction with our approach (see experiment above).  For example, if a training intervention is proposed to improve the fairness of LLM reasoning, then our FRM could be used to further monitor and improve the fairness of the model at inference time.
>
> If there are any specific techniques the reviewer believes are applicable to CoT-based inference, we would be grateful for the reference and happy to add more baselines.
>
> **[W1/Q3: Quality and bias of GPT-4o-mini annotations]**
>
> We appreciate the reviewer’s concern about label quality.  Our use of LLM-generated labels reflects a pragmatic approach common in recent work. Weak supervision is increasingly standard for tasks requiring large-scale, subjective annotation, especially when the ground truth is ill-defined. Our intent is to demonstrate a broader idea: that fairness can be modeled at the level of reasoning, not just outcomes, and that such an approach can yield fairness gains without sacrificing (or even improving) accuracy. Our FRM implementation is a proof of concept for this.
>
> We have updated our manuscript to explore the impact of using LLM-generated fairness data. We added a section discussing LLM annotation limitations and key failure modes from our human study. To give you an idea of this content, our qualitative study revealed a couple failure modes:
> 1. **Group names trigger biased labels:** The LLM may mark steps as biased where a group was mentioned even if the text is benign.
> 2. **Failure to recognize implicit bias:** The LLM may fail to label biased reasoning that is implicit and requires a deeper contextual understanding.
> 3. **Reasoning is incoherent, but LLM labels it as biased:** LLM annotates vague/incoherent reasoning as biased.
> 4. **LLM annotator believes hallucinated reasoning:** The model used for generating reasoning data occasionally hallucinated details based on stereotypes.
>
> Despite these annotation issues, the aggregate performance of our FRM suggests that the model is robust to some label noise. Future work could seek to improve annotation quality.
>
> We have also added a section with qualitative examples of FRM scores and outcomes to demonstrate how the weak-labeling approach leads to both successes and failures.  Finally, we note that our dataset will be released upon publication.
>
> **[W4: Fairness metrics as normalized improvement percentages]:**  We appreciate the suggestion to present the improvement percentages of our FRM and will add these results in the appendix.
>
> **[Q4: Fairness metrics and group fairness]:** While we agree that real-world fairness requires reasoning across multiple groups and considering different (sometimes context-specific) fairness metrics, our focus on binary-sensitive attribute settings and the equalized odds and equalized opportunity metrics follows standard practice in the literature:
> [Zemel et al., 2013](https://proceedings.mlr.press/v28/zemel13.html), [Hardt et al. 2016](https://arxiv.org/abs/1610.02413), [Agarwal et al., 2018](https://arxiv.org/abs/1803.02453), [Cruz & Hardt, 2024](https://arxiv.org/abs/2306.07261)
>
> We believe that this serves as a controlled and standard setting for evaluating reasoning-level bias mitigation.  We leave it to future work to consider how our framework can be applied to address important questions such as how to optimize for different notions of algorithmic bias and fairness.

---

> > ### Comment · Reviewer_yvMq · 2025-08-05
> > **Reponse to authors**
> >
> > Thank you for the detailed comments. I appreciate the authors’ efforts in addressing the concerns raised, particularly the inclusion of additional experimental results on fairness prompting baselines and the clarification regarding the scope and issues regarding the use of GPT 4o.
> >
> > I will consider your responses and other reviewers’ perspectives in determining my final score.

---

> > > ### Author Response · Authors · 2025-08-06
> > >
> > > Thank you for taking the time to review our responses. We are glad the additional experimental results and clarifications were helpful. As you finalize your decision, we are happy to answer any remaining questions.

---

### Official Review · Reviewer_wP53 · 2025-06-24

**Clarity:** 3
**Significance:** 2
**Originality:** 2
**Rating:** 4
**Confidence:** 3

**Summary:**

This paper proposes a method to enhance the fairness of large language models by evaluating their reasoning. First, the authors train a separate model to assign scores to individual reasoning steps. Then, at inference time, for each question they sample multiple chains-of-thought and apply the trained scorer to each step. The final answer is chosen via a weighted vote over these scored reasoning paths. In their empirical evaluation, the scoring model is trained on BBQ and tested on several other datasets.

**Questions:**

1. If the reasoning is not faithful, what specific issues would that lead to?
2. If we can acquire a reliable process reward model for fairness, would it be more effective to finetune the model with RL?

**Ethical Concerns:**

["NO or VERY MINOR ethics concerns only"]

**Final Justification:**

The authors have provided additional results that address my initial concerns. While I remain somewhat skeptical about the quality of the LLM-based judge, this limitation appears to be common these days and may not justify rejection on its own.

**Limitations:**

The authors acknowledge the limitation of their method, noting that using LLM-based judges might introduce additional bias.

**Quality:**

2

**Strengths And Weaknesses:**

LLMs are not my area of expertise, so I may have missed some related works or nuances in assessing the empirical study’s thoroughness.

## Strengths
1. Careful evaluation of the reasoning steps is a promising approach to improving fairness of LLMs.
2. The presentation is clear and the paper is easy to follow.

## Weaknesses
1. I am concerned about the quality of the trained reward model. Its training relies heavily on supervised learning, where label quality is crucial. However, they propose to use an LLM to generate these labels. It is unclear whether this step introduces bias. They attempt to evaluate this by having the authors label them and check alignment, but this is not convincing to me.
2. Regarding comparison to existing methods, the authors criticize prompt-based approaches as brittle, inconsistently followed, and prone to reducing output diversity. I am not convinced that the proposed method addresses these issues. For instance, as mentioned above, it relies on an external LLM, which may itself exhibit fairness problems. And it’s unclear why output diversity is relevant to the specific task under investigation.
3. The computational cost could be high, since each inference requires sampling multiple CoT and running a separate LLM to evaluate them. Is there any intuition on how many samples are sufficient for different tasks and models?
3. The empirical study is not convincing. I am not very familiar with the literature, but it seems they don’t include state-of-the-art baselines. They only compare to basic methods such as CoT and majority voting, which appear to be simple variants of existing LLM reasoning techniques. I wonder how it compares to existing work on LLM fairness—examples could be prompt-based methods like [1][2] or other post-training techniques like [3].

[1] Tamkin, Alex, et al. "Evaluating and mitigating discrimination in language model decisions." arXiv preprint arXiv:2312.03689 (2023).

[2] Li, Jingling, et al. "Prompting Fairness: Integrating Causality to Debias Large Language Models." The Thirteenth International Conference on Learning Representations.

[3] CFL: Causally Fair Language Models Through Token-level Attribute Controlled Generation

---

> ### Author Rebuttal · Authors · 2025-07-30
>
> We thank the reviewer for the time and consideration taken in reviewing our submission.  We are glad that they consider our approach to be promising, and that they found the paper to be clear and easy to follow.
>
> Below we respond to the individual concerns raised throughout the review.
>
> **[Weakness 1: Quality and bias of LLM annotations]**
>
> We appreciate the reviewer’s concern about label quality and discuss some of these concerns in our limitations section. Our use of LLM-generated labels reflects a pragmatic approach common in recent work. Weak supervision is increasingly standard for tasks requiring large-scale, subjective annotation, particularly in cases where the ground truth is ill-defined. Our intent is to demonstrate a broader idea: that fairness can be modeled at the level of reasoning, not just outcomes. The FRM is a proof of concept for this, and future work could improve on this method using higher quality supervised labels. In practice, our experiments show that even though the labels may be imperfect, they can still lead to significant downstream fairness gains.
>
> We have updated our manuscript to more clearly address this point. Specifically, we:
>
> - Added a section discussing the limitations of LLM annotations, examining each main failure mode observed in our human alignment study (a preview shown below):
>   - *Group-name triggers*: Labels sometimes flagged any mention of a group as biased, even when neutral.
>   - *Implicit bias missed*: The LLM failed to catch contextually embedded biases.
>   - *Misattributed bias*: Reasoning that was vague or incoherent was occasionally marked as biased.
>   - *Hallucinated details*: The reasoning model occasionally added stereotypical details not in the prompt, which the labeling model failed to flag.
>
> - Included additional qualitative examples of FRM scores and outcomes to illustrate both the strengths and limitations of our approach under weak supervision.
>
> - Committed to releasing our dataset of LLM labels upon publication, along with a dataset card (see Appendix) to support transparency.
>
> If the reviewer can specify what they found unconvincing about the human study we conducted to check the alignment of LLM labels with human judgments, we would be happy to speak to those concerns.
>
>
> **[Weakness 2: Comparison to prompt-based approaches]**
> Prompt-based approaches rely on the model following the prompt to improve fairness while still making accurate decisions. Given that LLMs cannot always follow prompts and respond to prompt changes in unpredictable ways ([Perez et al., 2021](https://arxiv.org/abs/2108.07258), [Webson & Pavlick, 2021](https://arxiv.org/abs/2109.01247), [Zhou et al., 2022](https://arxiv.org/abs/2212.09251)), in some cases this works, in other cases it doesn’t. We observed that in many cases, this can lead to the model following similar types of reasoning and giving the same answer every time to “play it safe” from a bias perspective. Our method addresses these issues by scoring reasoning steps. In this way, we don’t change the natural output diversity of the model and we provide transparency into which steps are downvoted as opposed to relying on the model to follow a fairness prompt and adjust its reasoning accordingly. Output diversity is important in these tasks because taking the majority vote over many reasoning chains has been widely shown to increase task accuracy, but if output diversity is significantly reduced, this can remove the benefits of a majority vote.
>
> - **New experiments**: To further address these concerns, we have performed additional experiments on a set of stronger fairness prompting baselines. Using the COMPAS task, we systematically evaluated 10 original fairness prompts (written by GPT-4o) alongside the 7 prompts proposed by [Tamkin et al., 2023](https://arxiv.org/abs/2312.03689) to reduce discrimination in high-stakes language model applications. Each prompt was tested with and without CoT and we additionally included a CoT@32 setting using the best-performing prompt from each group. To highlight the modularity of our approach, we also apply the FRM to combine the 32 CoTs produced using fairness prompts. We report equalized odds and equalized opportunity deviations, defined as the sum and individual differences in FPR and TPR across groups, consistent with our original paper.
>
> - **New findings**: Consistent with our hypothesis and original findings, some prompts yielded reductions in equalized odds and equalized opportunity. However, this came at the cost of significant reductions in accuracy. This tradeoff remained consistent across settings and accuracy was reduced further when we sampled 32 times with fairness prompts. In many cases, fairness was improved not through better reasoning but by predicting the same label for all inputs. These results reinforce our core claim that reasoning-level supervision offers a more effective and robust fairness intervention than prompting when accuracy is a primary concern. Our findings echo concerns in prior work, including Tamkin et al., regarding output distortion from fairness prompting.
>
>
> - **Results**:
>
> | Setting                        | Avg. Accuracy | Min. Accuracy | Max. Accuracy | Avg. Eq. Odds Dev | Avg. Eq. Opp Dev |
> |-------------------------------|---------------|----------------|----------------|----------------|----------------|
> | 10 fairness prompts           | 46.6          | 27.8           | 56.6           | 16.0            | 6.7            |
> | 10 fairness prompts w/ CoT    | 56.7          | 53.6           | 60.2           | 16.2           | 4.0            |
> | 7 Tamkin prompts w/ CoT       | 58.1          | 54.6           | 59.4           | 17.5           | 3.8            |
>
> **Best Settings**:
>
> | Setting                                   | Accuracy | Eq. Odds Dev | Eq. Opp Dev |
> |------------------------------------------|----------|----------|----------|
> | Best fairness prompt w/ CoT (majority@32) | 57.2     | 19.7     | 4.7      |
> | Best fairness prompt w/ CoT (FRM@32)      | 63.6     | 16.6     | 4.1      |
> | Best Tamkin prompt w/ CoT (majority@32)   | 56.0     | 4.7      | 0.5      |
> | Best Tamkin prompt w/ CoT (FRM@32)        | 56.0     | 5.7      | 0.3      |
> | **FRM (Ours)**                            | **65.3** | **15.3**  | **7.6**  |
>
>
>
> **[Weakness 3: Computational cost]**
>
> The goal in designing our algorithm was to facilitate fair decisions while incurring a minimal (or no) tradeoff with accuracy and without requiring any task-specific training data. This reflects:
>
>   - the practical need for accurate predictions
>   - legal realities around discrimination law in the US, where the existence of an equally accurate yet fairer model (i.e., a less discriminatory alternative) is a requirement for legal action against an organization accused of deploying discriminatory algorithms ([Black et al., 2024](https://arxiv.org/abs/2406.06817))
>
> Given the need for accuracy, our main baselines are those that sample multiple CoTs, and our FRM is further applied to this baseline. Thus, for a CoT inference example where *N* chains are sampled, the cost of applying our FRM only adds the compute related to batched inference to predict a binary label for each reasoning step in the *N* chains. This should require fewer FLOPS (and less wall clock time) than, e.g., autoregressively producing one more reasoning chain.
>
> **[Weakness 4: Baseline selection]**
>
> Our selection of baselines was driven by:
> 1. Accuracy as a first order concern, given practical and legal realities
> 2. Generalization to new decisions without further training data
>
> We are not aware of any existing fairness method that is directly applicable to CoT reasoning with many samples, which is the focus of our work given our constraint of achieving fairness under minimal disruptions to accuracy.
>
> **[Question 1: Impact of unfaithful reasoning]**
>
> If we understand this question correctly, the reviewer is wondering about the implications of the LLM ignoring its generated reasoning process when making the final decision.  We will aim to highlight this concern further in the limitations section, where we discuss how some reasoning steps may be inconsequential to the final decision.  We also highlight and analyze this issue in the previously mentioned sections exploring qualitative examples of the GPT-4 labels and test time labels and decisions.
>
> **[Question 2: Fine-tuning with RL]**
>
> This is a natural follow up step that we would be curious to see explored in future work. The benefit of our method, though, is that it can be applied out-of-the-box to any LLM and works with those that may be too unwieldy to train (particularly in an academic setting).  Also, applying our model at inference is in keeping with our focus on retaining prediction accuracy.

---

> > ### Comment · Reviewer_wP53 · 2025-08-05
> >
> > Thank you for the response.
> > 1. W1 and W2 - My concerns center on how much we can trust the reward model and how to systematically evaluate their quality. Regarding concerns about the human study, I am not fully convinced that "asking three of the authors of this paper to label a random sample of 100 reasoning steps each" is a scalable way of testing this. The authors criticize prompt-based methods as sometimes working and sometimes not. I don't see evidence that can prove this will not be the case for the proposed method.
> > 2. W4 - I don't understand why accuracy as a first concern would rule out prompt-based and post-training-based baselines such as those I suggested. Do we have any evidence that they are strictly worse than the proposed method in terms of accuracy such that no empirical justification is necessary?

---

> > > ### Author Response · Authors · 2025-08-06
> > >
> > > We thank the reviewer for their continued engagement with our work.  Please see our response to your remaining concerns below, and if there are any remaining issues we would be happy to work to clarify them.
> > >
> > > **[W1 and W2 - My concerns center on how much we can trust the reward model and how to systematically evaluate their quality. Regarding concerns about the human study, I am not fully convinced that "asking three of the authors of this paper to label a random sample of 100 reasoning steps each" is a scalable way of testing this. The authors criticize prompt-based methods as sometimes working and sometimes not. I don't see evidence that can prove this will not be the case for the proposed method.]**
> > >
> > > We believe that our use of an LLM for weak supervision is in keeping with standard practice, as is our evaluation suite [e.g., see [1],[2],[3],[4]].  As a useful example, we point to [1]. This paper presents one of the widely used Process Reward Models (PRM) to date, and our FRM is a type of PRM. They train a PRM using LLM labels and find that these labels generalize better than Monte Carlo synthesized ones and are competitive with human annotations. This is also the case in [2]. [3] performs a human study of a similar scope to ours, with 3 undergrad/grad students labeling.  They find that a GPT-4 judge exhibits a 59.7% agreement rate with human labellers in their case, compared to 75% in our study.
> > >
> > > While we believe that we introduce significant novelty in where and how we employ LLM supervision, the use of LLM supervision itself is quite common.
> > >
> > > [1] The Lessons of Developing Process Reward Models in Mathematical Reasoning
> > > https://arxiv.org/abs/2501.07301
> > >
> > > [2] RLAIF vs. RLHF: Scaling Reinforcement Learning from Human Feedback with AI Feedback https://arxiv.org/abs/2309.00267
> > >
> > > [3] UltraFeedback: Boosting Language Models with Scaled AI Feedback https://arxiv.org/abs/2310.01377
> > >
> > > [4] Judging LLM‑as‑a‑Judge with MT‑Bench and Chatbot Arena (Zheng et al., 2023) https://arxiv.org/abs/2306.05685
> > >
> > >
> > > **[W4 - I don't understand why accuracy as a first concern would rule out prompt-based and post-training-based baselines such as those I suggested. Do we have any evidence that they are strictly worse than the proposed method in terms of accuracy such that no empirical justification is necessary?]**
> > >
> > > In response to the reviewer’s concern about empirical evidence with respect to fairness prompting, our rebuttal included experimental results for 20+ new fairness prompting baselines. These included all 7 fairness prompts from the Tamkin et al. (2023) paper suggested by the reviewer, as well as stronger baselines combining fairness prompting with modern inference scaling techniques. Our method achieved 65.3% accuracy versus 57.2% and 56.0% for the best fairness and Tamkin prompts respectively.  If there was something specific about these experiments that felt insufficient or unclear, please let us know. However, we believe our original rebuttal addressed this concern, and also note that Reviewer yvMq seemed satisfied with these new results after having the same concern.  Finally, we would like to highlight that these results and observations are consistent with those in the Tamkin et al. (2023).
> > >
> > > With respect to the post-training or fine-tuning baselines, they are not applicable in our setting given that there is no training data available for the downstream tasks.  Making such distinctions when choosing baselines is standard practice in LLM research.  For example, in the Chain of Thought[5] and Tree of Thought[6] papers, their baselines are focused on prompting based methods.
> > >
> > > [5] Chain-of-Thought Prompting Elicits Reasoning in Large Language Models https://arxiv.org/abs/2201.11903
> > >
> > > [6] Tree of Thoughts: Deliberate Problem Solving with Large Language Models https://arxiv.org/abs/2305.10601

---

> > > > ### Comment · Reviewer_wP53 · 2025-08-07
> > > >
> > > > Thanks for your response. I have adjusted my overall rating accordingly: 2->4.
> > > >
> > > > **Summary**: The authors have provided additional results that address my initial concerns. While I remain somewhat skeptical about the quality of the LLM-based judge, this limitation appears to be common these days and may not justify rejection on its own.

---

> > > > > ### Author Response · Authors · 2025-08-07
> > > > >
> > > > > Thanks again for your time and feedback. We are pleased that the additional results addressed your initial concerns. In the final version, we will be sure to emphasize the limitations of the LLM labeler. Thank you for helping us make our paper stronger with your review.

---

### Official Review · Reviewer_z2Cx · 2025-07-01

**Clarity:** 2
**Significance:** 4
**Originality:** 4
**Rating:** 5
**Confidence:** 3

**Summary:**

This paper introduces the Fairness Reward Model (FRM), which reduces bias in the Chain of Thought (CoT) reasoning process. The FRM is trained using CoT examples that are generated with different LLaMa models and labeled with GPT-4o-mini. These examples are taken from the BBQ dataset. Experts from the author team validate the labels. The FRM is then used to create a fairness score for each step of a chain, which is aggregated into the chain's mean. For the final decision, these chain scores are converted into weights and used for a weighted vote among the candidates. To balance accuracy, they introduce a temperature parameter to control the impact of the fairness scores.

The FRM is evaluated on different tasks and LLMs to ensure generalization. These tasks are represented by various publicly available benchmark datasets. The performance (accuracy and fairness metrics) of the LLMs is compared with that of the FRM and several baselines. These baselines include CoT with one chain, CoT with a majority vote over 32 chains, and fairness prompting.

They found that the FRM improved fairness while maintaining accuracy comparable to the most effective baselines, with some even improving accuracy. The study also demonstrates the FRM's effectiveness with another LLM (Mistral) and the importance of its design choices in an ablation study.

**Questions:**

1.	Your contribution is referred to both as a framework and an algorithm (e.g., lines 32 and 38). Please clarify: is your primary contribution a conceptual framework, an algorithmic implementation, or both?

2.	The manuscript claims generalization of the FRM across LLMs and tasks. However, it is only evaluated on three tasks and two LLMs, one of which is also involved in generating training data. Clarify the limitations of this setup and rephrase your generalization claims accordingly.

3.	The datasets used in Sections 6.1 and 6.2 differ. Why weren’t the same datasets used across both evaluations? Additionally, Figures 4 and 5 are presented in inconsistent formats, making comparison difficult. Unify  the datasets and figures, or explain why they differ.

4.	You mention that temperature settings impact FRM performance. However, it remains unclear how this hyperparameter was chosen and whether it was tuned separately for each LLM. How can temperature be set in real-world use cases where no labeled test set is available? Did you apply a train/test split to tune it?

5.	Several statements lack citations (e.g., lines 18, 20, 24, 27, 31, etc.), and abbreviation usage is inconsistent. Revise the manuscript to ensure all claims are properly referenced and all abbreviations are clearly introduced and used consistently.

**Ethical Concerns:**

["NO or VERY MINOR ethics concerns only"]

**Final Justification:**

The authors have addressed my concerns and I am happy to recommend acceptance.

**Limitations:**

Yes

**Paper Formatting Concerns:**

Several statements lack citations (e.g., lines 18, 20, 24, 27, 31, etc.), and abbreviation usage is inconsistent. Revise the manuscript to ensure all claims are properly referenced and all abbreviations are clearly introduced and used consistently.

**Quality:**

3

**Strengths And Weaknesses:**

I appreciate the authors’ work on this important topic and find the proposed approach both relevant and promising. Overall, the paper is well-written and clearly structured. However, there remain several areas where the manuscript can be improved in terms of presentation and methodological rigor.

Presentation
-	Several statements throughout the paper lack citations (e.g.line 18, 20, 24, 27, 31, 42, 81, 89, 94). These should be supported by appropriate references.
-	When citing works in the text, the authors should include the names of the authors (e.g., line 154), rather than relying solely on numerical references.
-	The terminology should be used consistently. For instance, line 38 refers to the contribution as an algorithm, in line 32 it is a framework. The authors should clarify which term accurately describes their contribution.
-	Abbreviations must be introduced once and used consistently throughout the paper. There are several inconsistencies that may result from automated rewriting (e.g., using LLMs). I recommend the authors carefully review all abbreviations before submission (e.g., see lines 34, 36, 57, 69, 78, 101, 115, 242, 246).
-	The abbreviation in Figure 4 is not introduced correctly. Line 249 should refer to “FRM” instead.
-	The paper should also specify the train-test split used to train the FRM model.

Quality:
-	The paper makes three good contributions. However, their conclusions are too bold. The FRM is benchmarked on three different tasks. Which is good but does not enough to be sure that the FRM generalizes across a wide range of real-world tasks. The authors should stick to the results and present them as they are. The FRM generalizes across three different tasks represented by three different benchmark datasets.
-	The paper makes three valuable contributions. However, the conclusions drawn from the experimental results are overstated. While the FRM is evaluated on three different tasks (an encouraging start)-this is not sufficient to support a claim of generalization to a wide range of real-world tasks.
-	The authors should remain grounded in the empirical evidence and state that the FRM generalizes across three benchmark tasks, rather than generalizing more broadly. The model is tested on only two LLMs (LLaMA and Mistral), and one of them was also used to generate the training instances. This raises concerns regarding the strength of the generalization claims. While the results are promising, demonstrating generalization based on a single additional model is insufficient. The authors should present these positive results more cautiously, especially regarding generalization across LLMs.
-	The authors only applied the model to two different LLMs (LLaMa and Mistral). In addition one of the model was partly used for generating the training instances for the FRM. Again the conclusion that the FRM generalizes is to strong. The authors should present their positiv results even when switching the LLM for inference. Generalization is not shown with one more model.
-	The authors used different datasets for the experiment presented in 6.1 and 6.2. This raises two issues: 1) It is unclear why the same four datasets were not used consistently across both experiments, which weakens the claim of generalizability. 2) Figure 4 and Figure 5 use different formats, making comparisons unnecessarily difficult. This should be avoided.
-	What impact does the chosen temperature have? The authors mention in line 315 that the temperature impacts the performance of the FRM framework. If so, and if it depends on the LLM used for inference, they should discuss this more deeply, as it conflicts with generalization across different LLMs. How can the temperature be set without a test dataset? Did they use a training-testing split to determine it?

Significance and Originality:
-	The presented approach appears novel, and there are many ways to further develop it.

---

> ### Author Rebuttal · Authors · 2025-07-30
>
> We thank the reviewer for taking the time to review and offer feedback on our submission. Below we respond to the particular concerns as summarized in the questions section:
>
> **[Q1: Framework vs. algorithm terminology]**
>
> We understand that our somewhat interchangeable use of these terms may have caused some confusion.  With this work, we aim to contribute both a conceptual framework and an example algorithmic realization of that framework. Conceptually, we offer a framework for balancing accuracy and fairness in LLM decision-making by only intervening once the reasoning and decision generating processes have been completed.  We argue (and show empirically) that this offers flexible control over fairness/accuracy trade-offs that can be elusive using other existing approaches.
>
> To demonstrate the potential benefits of applying this framework, we propose one potential algorithmic instantiation of it in Section 3, and show how it can perform in practice via experiments.  Other instantiations could be imagined: for example, using strong human labels to train the FRM, or applying some weighted step combination based on estimated contribution to the final decision. In our final version of the paper, we will closely examine our usage of these terms and make sure they are appropriate to the context in which they are being used.
>
>
> **[Q2: Generalization claims]**
>
> **Across tasks**: We appreciate the reviewer’s suggestion to make sure we scope our claims to what can be supported by our experimental findings.  In response to this concern, we will modify the cited claim in line 270 to “across 3 real-world tasks” (removing “a wide range of”).  We will also closely check the paper for any further unsubstantiated claims (although we note that we identify no further such instances in our preliminary check).
>
>
> **Across LLMs**: We appreciate the reviewer’s feedback that more exploration of transfer across reasoning models could further strengthen experiments. Distribution shifts may come from many factors, and thus it can be hard to thoroughly test for generalization. In this work, we chose to focus more on generalization across tasks and protected attributes than models.  This is because, practically, it is easier to train one FRM per reasoning model (using that model in step 1, section 3.1) than it would be to train one FRM per task and/or group, given that most organizations will serve only a handful of LLMs, but these LLMs may face a diverse set of decisions concerning people from many different groups, and often no relevant data is available. In response to this concern, we will modify the paper to clarify the limitations of our results on LLM generalization, and also highlight that we believe the other dimensions of generalization (tasks and groups) are of more concern.
>
> **[Q3: Experimental design clarifications]**
>
> **Dataset usage in different sections**: We used our full set of 4 (task/dataset, attribute) pairs for our main experiment in Section 6.1. For the rest of our experiments, we included as many results as possible given our goals and the constraints on our academic compute budget and space in our submission. In Section 6.1, we aimed to test the FRM across a diverse set of tasks and demographic contexts; in Section 6.2, our focus was on model generalization, specifically to reasoning chains generated by a previously unseen LLM (Mistral).  While we hope our work and open source asset releases spur further empirical research in this area, we believe that our experiments stand as a solid foundation for building on this approach to fair decision-making.
>
> **Figure formatting**: The rows and columns in Figures 4 and 5 were flipped to save space in our submission. In the final version, we can unify the format for the figures given the extra space.
>
>
> **[Q4: Temperature parameter]**
>
> The temperature parameter is meant to control the emphasis on accuracy versus fairness in combining the final decisions.  As this is a key hyperparameter in our algorithm (and the only one that needs to be chosen given a baseline of CoT w/ majority voting), we ablate its effect across three tasks and protected attributes in Figure 9, which is in appendix C because of space constraints. We believe that our ablation results indicate that this is a promising direction for flexible control of these key tradeoffs.
>
> For each LLM tested, we set the temperature via a small amount of hand tuning (comparing among 0.01, 0.2, 0.4, and 0.8) on a single task.  Since for a given LLM, a single temperature works across different tasks, this level of tuning should be possible in practice.  Finally, we note that our method does not “fail” under any temperature setting in the ablation, it’s just a matter of choosing the desired tradeoff level.  We will clarify this in the experiments section of our main paper. Additionally, the train-test split we use is included in Section 3.1. As noted in the paper, our dataset will be released upon publication to further support reproducibility and research.
>
> **[Q5: Editorial issues]**
>
> We thank the reviewer for carefully reading our submission and highlighting several specific editing issues. We will correct these issues in the final camera ready version of our work.

---

> > ### Comment · Reviewer_z2Cx · 2025-08-05
> >
> > The authors clarified many of my points raised — the points on presentation / reference adding still remains unclear to me.

---

### Official Review · Reviewer_jRkp · 2025-07-03

**Clarity:** 4
**Significance:** 4
**Originality:** 4
**Rating:** 5
**Confidence:** 5

**Summary:**

This paper proposes a method for improving fairness in large language model (LLM) decision-making by leveraging Chain-of-Thought (CoT) reasoning traces and a Fairness Reward Model (FRM). The FRM is trained to score CoT traces as "fair" or "unfair" using synthetic supervision signals based on heuristic labeling. At inference time, the LLM generates multiple CoTs per input; the one with the highest fairness score (as judged by the FRM) is used to produce the final output.
The authors evaluate this approach on several tasks involving morally or socially sensitive decisions that include hiring judgments, moral reasoning, and toxic span detection. The method consistently reduces disparities in outcomes across demographic groups without significant loss in task performance.

**Questions:**

1. The authors' method implicitly optimizes for demographic parity. How would it generalize to other fairness notions like equalized odds, calibration, or even counterfactual fairness? Could the FRM be adapted to alternative definitions?
2. Have you observed cases where reranking based on FRM leads to worse outputs? Particularly, outputs that are more ambiguous or inconsistent with prompt intent? If so, in what cases has this happened and can this be mitigated through prompting alone?

**Ethical Concerns:**

["NO or VERY MINOR ethics concerns only"]

**Limitations:**

I am most concerned about the limitation around the normative ambiguity in fairness definitions. Demographic parity may be inappropriate or even harmful in some contexts and as I explain above may not be the most robust way to mitigate fairness concerns. I am also concerned about he fragility of the FRM- particularly the potential of a pipeline that is sensitive to annotation artifacts or overfit to superficial cues. I worry about the ability to scale or adapt in production environments.

I will place this paper as a accept for now, but if my concerns can be adequately addressed by the authors, I would easily move this paper to a strong accept. Outside of concerns about the above, I think this is a very strong paper.

**Quality:**

4

**Strengths And Weaknesses:**

I have some thoughts that I will format based on specific dimensions:

1. Quality
- Strength: The paper is technically sound and proposes a modular, zero-shot-compatible fairness method that works without fine-tuning the base model. The reuse of CoTs as intervention points is clever and grounded in recent work on interpretability and LLM internal reasoning and goes through the appropriate lengths to give credence to seminal work in the space (Wei et al's CoT paper).
- Weakness: The fairness model relies on heuristically labeled training data to define what constitutes a "fair" CoT. I worry this introduces the risks of reflective bias, where the FRM simply learns the annotators’ normative assumptions rather than identifying universally unfair reasoning.

2. Clarity:
- Strength: The overall writing is clear and well-structured
- Weakness: The distinction between “fair reasoning” and “fair outcomes” is underexplored. Also, I am not always clear on whether the FRM is improving final outcomes, internal logic, or both when reading the paper.

3. Significance:
- Strength: This is an important contribution in the context of black-box LLM fairness, where fine-tuning or RLHF is often infeasible. The framework offers a tractable way to influence fairness using only CoTs and inference-time reranking.
- Weakness: The broader impact is limited by key assumptions: demographic parity is treated as a proxy for fairness, and the causal structure of decisions is assumed but not modeled. As with many fairness interventions, performance gains may mask deeper issues of epistemic validity. One thing of note is that, in the book Fairness and Machine Learning by Barocas et al (Chapter 8), they speak to different types of discrimination: structural, organizational, and interpersonal. These can be divided even further to two different tiers: direct and indirect). The FRM approach, by optimizing for demographic parity without accounting for how these layers of discrimination interact, risks targeting surface-level disparities while ignoring deeper structural harms. . Without formal modeling of the generative process behind both data and decision-making, there's a risk that the model simply learns to “look fair” without substantively mitigating harm.

4. Originality:
- Strength: Repurposing CoTs for fairness intervention is novel. The idea of scoring internal reasoning rather than final output is directionally aligned with interpretability and agent alignment goals.
-Weakness: None.

---

> ### Author Rebuttal · Authors · 2025-07-30
>
> We thank the reviewer for their thoughtful and positive feedback! We are encouraged that they found our Fairness Reward Model (FRM) to be novel, an important contribution, and highly generalizable. We respond to the particular concerns raised below:
>
> **[Q1: Risk of Reflective Bias]**
>
> Excellent question! We agree that any attempt to codify fairness is inherently normative. Our goal, however, is not to identify universally unfair reasoning but to make our assumptions explicit and grounded in prior empirical findings. Specifically, our heuristic is based on prior work demonstrating that CoT applied to socially sensitive contexts can systematically amplify stereotypical associations ([Shaikh, 2023](https://arxiv.org/abs/2212.08061)). We believe that it is better to specifically name one’s fairness priors, however contingent, than to use the ones embedded deeper in the model.
>
> The FRM is additionally constructed to be flexible: it can be retrained using a different fairness heuristic for another deployment setting. Our ultimate goal is to reduce bias on downstream tasks as measured by established fairness metrics. Our positive results on these tasks demonstrate that, while the labels may carry some normative assumptions, they are universal enough that they effectively reduce unfair results on objective downstream metrics.
>
>
> **[Q2: Fair Reasoning vs. Fair Outcomes]**
>
> Our results show both improved final outcomes (improved fairness while maintaining accuracy) and improved reasoning (down-weighting the contribution of unfair reasoning to the final outcome). Fair reasoning does not guarantee a fair outcome in every case; however, it increases the likelihood of fair outcomes and makes those outcomes more interpretable. Even in cases where outcome disparities persist, we think fair reasoning is an intrinsically valuable objective.
>
> We will make this process-outcome distinction more explicit and we have added qualitative examples to the paper that highlight these edge cases. For example, we have included an instance where the model reasons fairly but chooses an outcome that contradicts that fairness.
>
>
> **[Q3: Surface-Level Disparities vs. Deeper Structural Harms]**
>
> We recognize the concern that a model trained to reward fairness in its reasoning may be susceptible to targeting surface-level disparities. The FRM is not designed to appear fair for the sake of appearances, instead, it is trained to disincentivize reasoning that relies on stereotypes  or irrelevant demographic information. For example, a CoT that justifies its decision by referencing someone’s race when it is not relevant would receive a low score regardless of whether the final answer appears fair. We do not claim to solve all types of bias but we do believe that mitigating bias in reasoning is an important objective. We see process-supervision as a complementary strategy to interventions that operate at other levels such as data augmentation and causal modeling.
>
> **[Q4: Generalization to other fairness notions]**
>
> We share the view that generalizing to different fairness definitions is critical. We would like to emphasize that our method does not optimize for equalized opportunity and/or equalized odds. We use these only as evaluation metrics, based on their prominence in the ML fairness literature.  Our algorithm should be able to be adapted to improve outcomes according to most popular fairness definitions of machine learning fairness.  We see this adaptability as a strength of the approach of using process-level fairness supervision. We thank the reviewer for highlighting this important issue, and we will update our final paper to emphasize this point.
>
> **[Q5: Cases where FRM leads to worse outputs]**
>
> FRM scores only reflect fairness, which we agree is not sufficient for selecting the highest quality outputs overall. To address this, we introduce a temperature parameter that balances optimization for accuracy with the FRM score. One might also imagine other ways for integrating our FRM score into final output selection, for example considering it alongside some reward model in a chatbot setting.
>
> **[Q6: Limitations regarding demographic parity]**
>
> We agree that demographic parity is not universally appropriate and we wish to clarify that the FRM does not explicitly enforce any particular fairness metric. Demographic parity is a standard evaluation metric for ML fairness that we used to demonstrate the FRM’s impact on downstream performance. The FRM is agnostic to how fairness is measured.

---

> > ### Author Response · Authors · 2025-08-06
> > **Discussion period ending soon**
> >
> > Hello! Thank you again for taking the time to share such a thoughtful review. While we completely understand that it is a busy time for everyone, we wanted to follow up and see if we have addressed your concerns or if there are any further questions we can answer. The discussion period will be ending soon (August 8), and would love the opportunity to clarify anything further if needed. Thank you!

---

> > ### Comment · Reviewer_jRkp · 2025-08-09
> >
> > Thank you for the detailed rebuttal. Here are my thoughts on your responses:
> >
> > Risk of Reflective Bias:
> > While I appreciate your acknowledgment of the normative nature of fairness and the grounding of your approach in prior work, the use of heuristic labeling still carries the risk of reinforcing existing biases. The flexibility to retrain the FRM in different contexts is valuable, but it would help to clarify how you can mitigate biases when adapting to new settings, especially in high-stakes domains.
> >
> > Fair Reasoning vs. Fair Outcomes:
> > The distinction between fair reasoning and fair outcomes is important, yet it requires more clarity in the paper. As you note, fair reasoning increases the likelihood of fair outcomes, but it doesn't guarantee them. It would strengthen the paper to explicitly address situations where fair reasoning may lead to outcomes that appear inconsistent with fairness.
> >
> > Surface-Level Disparities vs. Deeper Structural Harms:
> > While the FRM focuses on reducing reasoning based on irrelevant demographic information, the emphasis on demographic parity risks addressing only surface-level disparities. It would be beneficial to explore how the FRM can be extended or integrated with broader fairness frameworks that address deeper, structural forms of discrimination, such as organizational or interpersonal biases.
> >
> > Generalization to Other Fairness Notions:
> > The potential to adapt the FRM to other fairness metrics, such as equalized odds or counterfactual fairness, is a notable strength. More detail is needed, though, on how the FRM would handle these definitions in practice. It would be helpful to discuss the challenges and trade-offs involved in adapting to different fairness metrics.
> >
> > Cases Where FRM Leads to Worse Outputs:
> > Introducing a temperature parameter to balance fairness and accuracy is a practical solution. That said, further exploration is needed on situations where reranking based on fairness could degrade output quality, particularly in cases where context matters. A clearer explanation of how you can prevent this degradation would increase confidence in the approach.
> >
> > Limitations Regarding Demographic Parity:
> > I appreciate your clarification that the FRM does not explicitly enforce demographic parity but uses it as a standard evaluation metric. Still, demographic parity may not always be the most appropriate fairness metric. A deeper discussion on its limitations, and how the FRM can be applied in contexts where it isn’t suitable, would strengthen the paper’s broader applicability.
> >
> > I am going to maintain the Accept rating and look forward to seeing this paper be presented. Wonderful work.

---

### Comment · Area_Chair_q9zH · 2025-08-05
**Paper discussion**

Dear reviewers,

Thank you for your thoughtful comments!

If you haven't done so, please take time to check the author's responses. Please note that you are required to formally respond to the author's rebuttal before submitting the "Mandatory Acknowledgement". Irresponsible reviewers will be flagged.

Thanks,
Your AC

---

### Note · Authors · 2025-08-14

We thank the reviewers for their thoughtful and helpful comments. We are glad the reviewers recognized the novelty, generalizability, and clarity of our approach, and that we were able to address concerns about prompting baselines and the use of LLM labelers. Our final version will incorporate all of the changes requested by reviewers including the additional experiment results and a section on limitations of our weak-labeling strategy.

---

### Decision · Program_Chairs · 2025-09-17

**Decision:**

Accept (poster)

**Comment:**

This paper introduces the Fairness Reward Model to improve the fairness of the CoT reasoning process. Overall, the proposed framework is technically solid and has been tested with comprehensive evaluations. The reviewers unanimously agree that this paper should be accepted.